# Exosome-delivered EGFR regulates liver microenvironment to promote gastric cancer liver metastasis

Haiyang Zhang[1], Ting Deng[1], Rui Liu[1], Ming Bai[1], Likun Zhou[1], Xia Wang[1], Shuang Li[1], Xinyi Wang[1], Haiou Yang[1], Jialu Li[2], Tao Ning[1], Dingzhi Huang[1], Hongli Li[1], Le Zhang[1], Guoguang Ying[1] & Yi Ba[1]

The metastatic organotropism has been one of the cancer's greatest mysteries since the 'seed and soil' hypothesis. Although the role of EGFR in cancer cells is well studied, the effects of secreted EGFR transported by exosomes are less understood. Here we show that EGFR in exosomes secreted from gastric cancer cells can be delivered into the liver and is integrated on the plasma membrane of liver stromal cells. The translocated EGFR is proved to effectively activate hepatocyte growth factor (HGF) by suppressing miR-26a/b expression. Moreover, the upregulated paracrine HGF, which binds the c-MET receptor on the migrated cancer cells, provides fertile 'soil' for the 'seed', facilitating the landing and proliferation of metastatic cancer cells. Thus, we propose that EGFR-containing exosomes derived from cancer cells could favour the development of a liver-like microenvironment promoting liver-specific metastasis.

[1] Tianjin Medical University Cancer Institute and Hospital, Tianjin Key Laboratory of Cancer Prevention and Therapy, National Clinical Research Center for Cancer, Tianjin 300060, China. [2] Department of Gastroenterology, Tianjin First Center Hospital, Tianjin 300192, China. Correspondence and requests for materials should be addressed to G.Y. (email: yingguoguang163@163.com) or to Y.B. (email: bayi@tjmuch.com).

The fact that certain tumours are inclined to metastasize to specific organs has been recognized for over a century[1]. The Paget's 'seed and soil' hypothesis suggests that the successful growth of metastatic cancer cells largely depends on the properties of target organs (soil) and cancer cells (seeds)[1,2]. Liver is the organ where various types of metastatic tumours take place[3,4]; however, the knowledge on the mechanism that liver promotes cancer cell colonization and growth is still absent.

Exosomes are small vesicles that are secreted from cells and have been found to mediate signalling transduction between neighbouring or distant cells[5,6]. Exosomes (30–200 nm) and shedding vesicles (200–1,000 nm) are two main forms of extracellular vesicles. The previous study has shown that exosomes bear surface receptors or ligands of the original cells; therefore, they have the tendency to specifically interact with target cells[7]. Although exosomes are well known to deliver microRNAs (miRNAs) and messenger RNAs[8–10], the role of proteins in exosomes, especially membrane proteins, has not been fully understood yet. Epidermal growth factor receptor (EGFR) is located in the cytomembrane, which is well known to play a dominant role in tumorigenesis and development. Recent studies showed that EGFR can be secreted from cells via the transport of vesicles and these EGFR-containing exosomes are proved to regulate signalling pathways of endothelial cells and T cells[11–13]. Moreover, microvesicles containing EGFRvIII are found to merge with the plasma membranes of cancer cells lacking this type of receptor and the share of EGFR mutants between cancer cells promote tumour development[14].

Hepatocyte growth factor (HGF) was first discovered in mouse liver and has been found to be linked with tumour development. Serum HGF is upregulated in various types of cancer, which is a potential biomarker for prognosis[15–17]. C-MET is the receptor of HGF and is widely expressed in various types of cancer. The HGF-cMET pathway is involved in cell invasion, proliferation and angiogenesis, and is believed to be a novel target for cancer therapy[18,19]. Gastric cancer (GC) with liver metastasis is one of the main forms in advanced GC[20,21]; however, the molecular mechanism in this process remains unclear. Liver has adequate supply of blood and may provide nutrition for cancer cells; however, the role of paracrine growth factors has not been the cause for concern. Liver-derived HGF may contribute to the landing and fast growth of metastatic GC cells.

In the present study, we first find that c-MET, but not HGF, is highly expressed in the liver metastases of GC, suggesting that GC metastases mainly bind with liver paracrine HGF. Exosomes, derived from GC cells, are proved to activate liver HGF by suppressing miR-26a and miR-26b; the two miRNAs directly target the 3'-untranslated region (UTR) of HGF mRNA. Subsequently, we show that secreted EGFR, which is found in the exosomes of GC serum and GC cells, is finally located in membrane of mixed liver cells, including stromal cells. In addition, EGFR-absent exosomes lost the ability to regulate miR-26/HGF pathway in the liver. Moreover, in vivo studies provide direct evidence that liver HGF plays a key role in determining the ratio of hepatotropic metastasis as well as the growth of liver metastases. Hence, exosomes secreted from primary gastric tumour regulate liver micro-environment to promote liver metastasis and the upregulated liver paracrine HGF provides fertile 'soil' for the metastatic cancer cells.

## Results

### EGFR is located in the serum exosomes of GC.
Although EGFR is well known to be upregulated in tumour tissues, few studies have been focused on circulatory EGFR delivered by exosomes. We first isolated serum exosomes (sr-exosomes) by high-speed centrifugation and determined EGFR levels. As is shown in Fig. 1a, the sizes of these exosomes were mostly around 100 nm.

EGFR is enriched in sr-exosomes of GC patients but not in exosomes of normal human serum ($n = 20$); full-length EGFR was detected at 185 kDa (Fig. 1b). In addition, the content of exosome EGFR was increased in serum of stage IV GC patients ($n = 20$, Fig. 1b). These results illustrated that GC sr-exosomes contains EGFR oncoprotein, which may play an important role in the development of GC.

### The expression of HGF and c-MET in GC liver metastases.
Although HGF has been reported to be upregulated in various types of cancer, the expression pattern of HGF in tumour metastases is little known. To explore whether HGF is expressed in the liver metastases of GC, we determined HGF expression by using immunohistochemistry and western blotting. The results showed that HGF is highly expressed in para-carcinoma tissues and liver but not in the GC metastases (Fig. 1c,d). However, the HGF receptor, c-MET and phosphorylated c-MET (p-c-MET) is obviously expressed in GC liver metastases (Fig. 1c). Liver metastases from 30 patients were detected and the positive detection rate of HGF is only 15%, whereas the positive rate of c-MET is 90% (Fig. 1e). These data suggested that the cancer cells in the metastases mainly bind with liver HGF, which is released into the liver microenvironment through paracrine manner.

### Exosome-mediated EGFR is located in the liver cell membrane.
Although EGFR is found in the sr-exosomes of GC patients, the levels of EGFR in exosomes secreted from GC cells is not known. As EGFR is known to be a membrane protein, the correct location of exosome-mediated EGFR in target cells is important for it to function. In this study, exosome secreted from SGC7901 cells (SGC-exosomes) were isolated (Fig. 2a) and co-cultured with primary mouse liver cells. This indicated that the primary liver cells contain stromal cells, such as kupffer cells. We first checked the exosome EGFR levels of SGC7901 cells. As is shown in Fig. 2b, EGFR was detected in the exosomes of SGC7901 cells, whereas the levels of EGFR in both SGC cells and exosomes were strongly decreased by transfection of small interfering RNA (siRNA). To confirm the biological function of exosome EGFR, $5 \times 10^5$ primary mouse liver cells were incubated with 50 μg exosomes derived from SGC7901 cells (Fig. 2c). These exosomes were found to rapidly enter into the recipient cells (Fig. 2d). Moreover, green fluorescent protein (GFP)-tagged EGFR was expressed in SGC7901 cells and the SGC-exosomes were isolated and incubated with mixed primary liver cells. As is expected, the GFP-tagged EGFR is detected in the outer membrane of mixed liver cells and is co-localized with E-cadherin (Fig. 2e). To monitor the concentration and sizes of exosomes secreted by SGC7901 cells, Nanosight NS300 system was used (Fig. 2f). Most of the isolated vesicles were around 100 nm, which is the typical size of exosomes. Taken together, these results clearly demonstrated that the membrane protein EGFR can be delivered into the liver by SGC-exosomes and implied that EGFR is located in the membrane system through membrane fusion.

### Characterization of primary liver cells.
The types of primary liver cells were characterized using marker stromal cells and liver cells. As is shown in Fig. 3a,b, F4/80 (marker for Kupffer cells), alpha smooth muscle actin (α-SMA) and desmin (markers for hepatic stellate cells) are enriched in primary cells. Theses markers were also detected by using immuno fluorescence (Fig. 3c) and it is proved that SGC exosomes containing EGFR-GFP can be taken up by Kupffer cells and hepatic stellate cells (Fig. 3c).

### Exosome EGFR activates liver HGF by suppressing miR-26a/b.
HGF is mainly expressed in the liver and is known to be a tumour

promoter[22]. Therefore, we subsequently checked the effects of SGC-exosomes on liver HGF. Forty micrograms of SGC exosomes were added into the medium of $10^6$ primary liver cells seeded in a six-well plate. It was showed that SGC-exosomes significantly promote HGF and EGFR expression in mixed liver cells (Fig. 4a); however, enhanced HGF expression was blocked when EGFR was removed from exosomes (Fig. 4a). Similarly, overexpression of EGFR in primary liver cells results in the upregulation of liver HGF (Fig. 4c). However, both SGC-exosomes and EGFR overexpression cause little change of HGF mRNA (Fig. 4b,d). Thus, it is believed that exosome

EGFR activates liver HGF by suppressing its upstream miRNAs. Among all the HGF-related miRNAs predicted by bioinformatics methods, miR-26a and miR-26b were found to be downregulated in GC (Fig. 4e,h). In addition, both SGC-exosomes and overexpressed EGFR strongly inhibit miR-26a/b expression in liver cells (Fig. 4f,g). Therefore, exosome-mediated EGFR regulates HGF expression by suppressing miR-26a/b in liver cells.

**miR-26a/b directly target HGF in the liver.** To give direct evidence of the interaction between miR-26a/b and HGF, we used

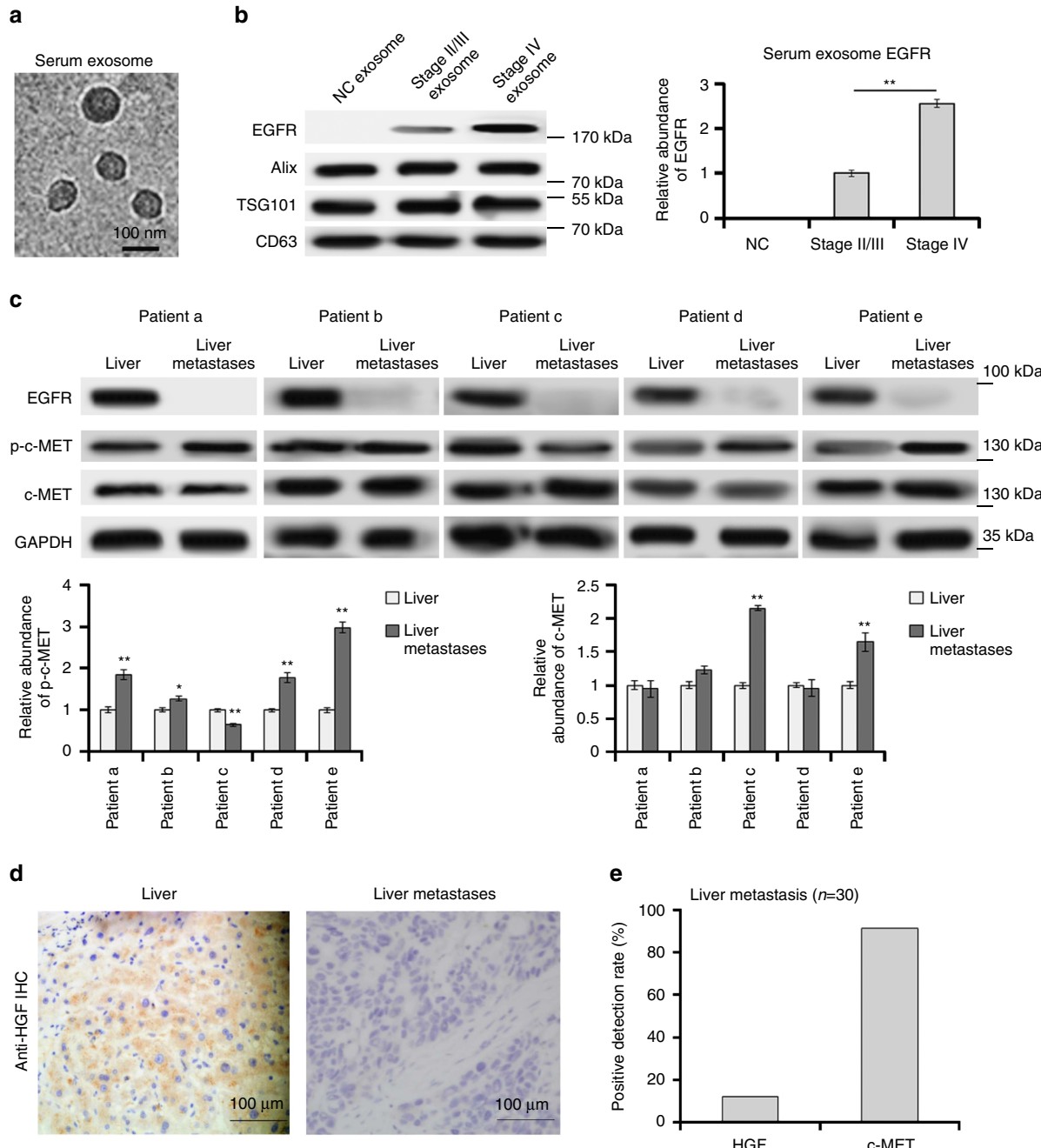

**Figure 1 | Clinical analysis of sr-exosome EGFR and HGF-cMET in GC.** (**a**) Electron microscope scanning of exosomes isolated from human serum. (**b**) Sr-exosome EGFR is related to the progression of GC. Exosomes were isolated from serum of healthy donors (NC), stage II/III GC patients and stage IV GC patients, respectively ($n = 20$); Alix, TSG101 and CD63 were used as the internal control of exosomes. (**c**) The expression of HGF, c-MET and p-c-MET in para-carcinoma tissue (liver) and GC liver metastases ($n = 5$). (**d**) Immunohistochemistry (IHC) analysis of HGF in GC liver metastasis. (**e**) Positive ratio of HGF and its receptor c-MET in GC liver metastases ($n = 30$). The data represent the mean ± s.e.m. *$P < 0.05$, **$P < 0.01$ (Student's $t$-test).

luciferase reporter plasmid containing either wild-type or mutant 3′-UTR of HGF mRNA; the binding sites of miR-26a/b were show in Fig. 5a. It was shown that the luciferase activity was markedly reduced in the cells overexpressed miR-26a or miR-26b, whereas the inhibition of miR-26a/b relatively enhanced luciferase activity (Fig. 5b). However, the inhibitory activity of miR-26a/b on luciferase activity was lost when the binding sites were lost (Fig. 5b). The HGF protein and mRNA were also assessed in primary liver cells transfected with miR-26a/b mimics or inhibitors. Western blotting analysis revealed that HGF expression was strongly suppressed by over-expressed miR-26a or miR-26b, whereas the inhibitors of miR-26a/b relatively enhanced HGF expression (Fig. 5c). miRNAs are well known to suppress gene expression at the posttranscriptional level and, as expected, HGF mRNA remained unchanged with the effects of miR-26a/b mimics or inhibitors (Fig. 5d). In conclusion, miR-26a and miR-26b regulate HGF expression in primary liver cells by directly targeting the 3′-UTR of HGF mRNA.

**HGF regulates biological behaviour of GC cells *in vitro*.** Next, we assessed the effects of liver HGF on the promotion of cell invasion and migration of GC cells. Primary liver cells were divided into two groups. One group was treated with lenti-virus-containing HGF-overexpressing sequence or HGF short hairpin RNA (shRNA); the other group was treated with SGC exosomes or 293T exosomes. These primary liver cells were indirectly

co-cultured with SGC7901 cells using the 0.4 μm polyester membrane (Fig. 6a); the liver-secreted factors, such as HGF, can pass through the membrane freely. We use the indirect co-culture system to simulate the microenvironment of the liver. HGF expression and secretion in liver cells was determined by western blotting and enzyme-linked immunosorbent assay (ELISA) assay, respectively. It was showed that HGF-overexpressing lentivirus and SGC-exosomes significantly enhanced liver HGF expression and release (Fig. 6b,d). Exosomes of SGC7901 cells were also found to clearly promote HGF secretion (Fig. 6c). Subsequently, the effects of liver HGF on biological behaviour of SGC7901 cells were determined by EDU assay and Transwell assay, respectively. As is expected, primary liver cells overexpressed HGF and treated with SGC-exosomes strongly promoted proliferation, migration and invasion of SGC7901 cells, whereas the liver cells treated with HGF shRNA play the opposite function (Fig. 6e–g). As is expected, p-c-MET was increased with HGF (Fig. 6h). These *in vitro* results support that liver HGF plays a key role in regulating biological behaviour of GC cells.

***In vivo* role of liver HGF in liver metastasis.** To access the *in vivo* effects of liver HGF on the formation and growth of liver metastases, we subsequently established mouse tumour model. Before the orthotropic implantation of GC, mouse livers were treated with lentivirus containing either HGF-overexpressing sequence or HGF shRNA by multi-point injection. Mice were

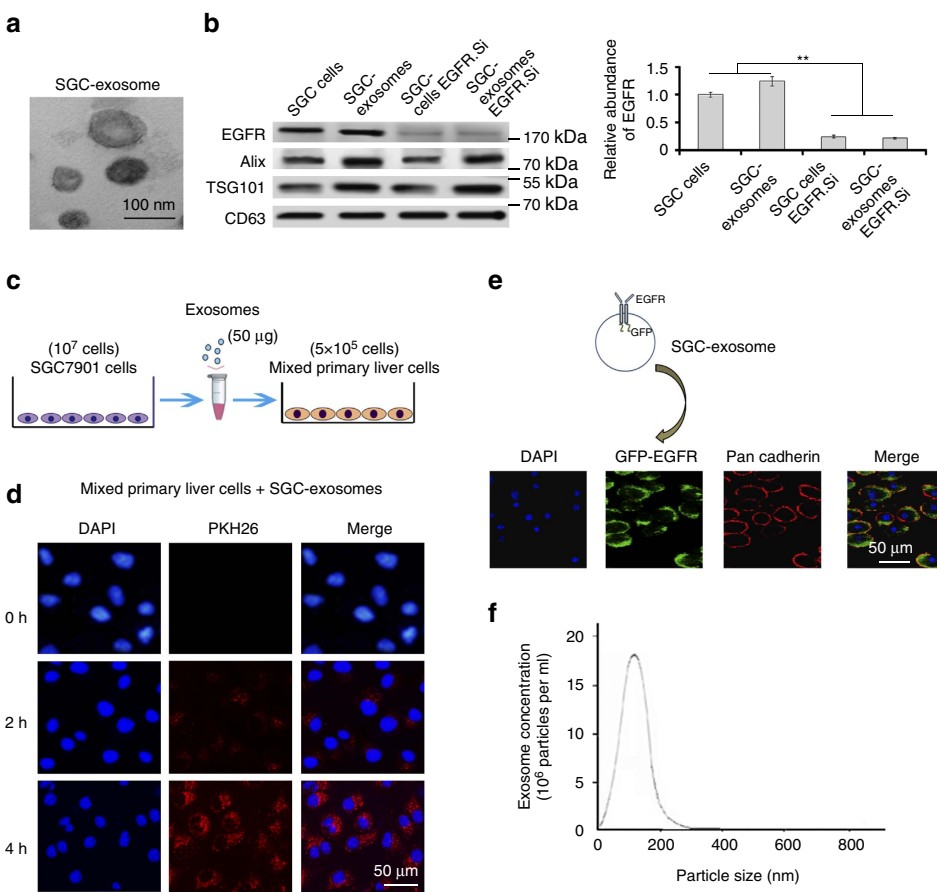

**Figure 2 | SGC-exosomes transport EGFR into liver cells. (a)** Electron microscope scanning of exosomes isolated from the medium of SGC7901 cells. **(b)** EGFR expression in both SGC exosomes and SGC7901 cells treated with EGFR siRNA. **(c)** Schematic description of the experimental design. The SGC exosomes were isolated and 50 μg exosomes were used to culture with $5 \times 10^5$ primary liver cells. **(d)** Confocal microscopy image of the internalization of fluorescently labelled exosomes in mixed liver cells. Scale bars, 50 μm. **(e)** SGC-exosome-mediated EGFR is located in the membrane of mixed primary liver cells. Scale bars, 50 μm. **(f)** Nanoparticle Tracking Analysis (NTA) of isolated exosomes. The data represent the mean ± s.e.m. **$P < 0.01$ (Student's *t*-test).

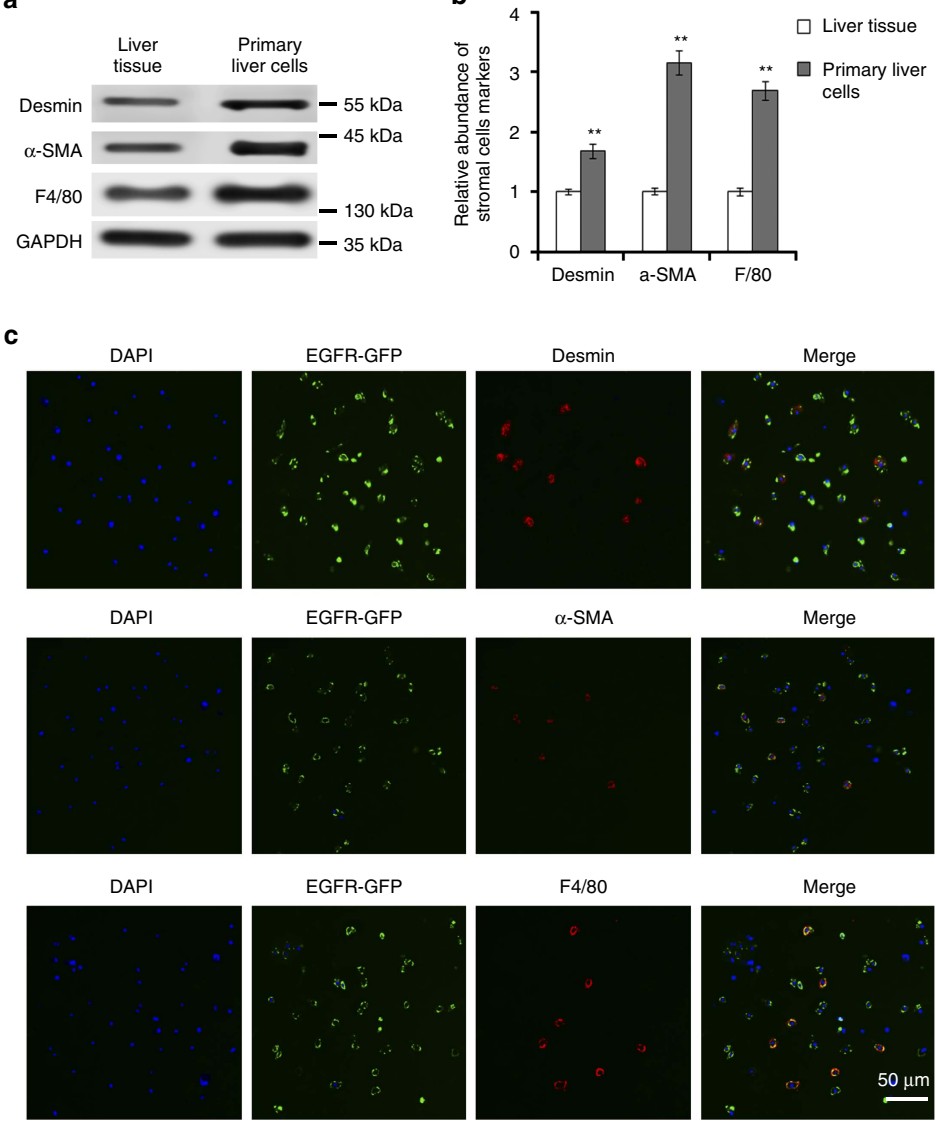

**Figure 3 | Characterization of cells types in primary liver cells.** (**a**) Western blotting analysis of desmin, α-SMA and F4/80 in mixed primary liver cells ($n = 3$). (**b**) Quantification analysis of **a**. (**c**) Immunofluorescence analysis of exosome-delivered EGFR and stromal cell markers. Scale bars, 50 μm. The data represent the mean ± s.e.m. **$P < 0.01$ (Student's $t$-test).

killed on the 66th day; mouse liver, serum, primary tumour and liver metastases were obtained on the 60th day post tumour implantation, followed by the recording of tumour metastases and gene expression.

The flowchart of *in vivo* experimental design is shown in Fig. 7a. EGFR in the primary tumours were first detected and it is overexpressed in the tumour tissues (Fig. 7b). Sr-exosomes of tumour-implanted mice were isolated (Fig. 7e) and EGFR is highly expressed in the sr-exosomes of tumour-implanted mice, but not in the control group (Fig. 7c). As is expected, liver HGF is significantly upregulated with the implantation of tumour; multi-point injection of lenti-virus containing HGF shRNA significantly suppressed HGF expression in the liver (Fig. 7d). EGFR expression was also clearly increased in the liver as a result of tumour implantation (Fig. 7d). Moreover, the high levels of exosome-EGFR in mouse serum resulted in the inhibition of miR-26a/b (Fig. 7f).

These data based on tumour xenograft model further demonstrated that tumour-secreted exosomes activate liver HGF though suppressing miR-26a/b expression.

**Exosome-EGFR is located in liver cytomembrane *in vivo*.** SGC7901 cells were overexpressed with EGFR-GFP and exosomes were isolated and injected into mice through the tail vein. Location of exosome EGFR was determined using anti-GFP antibody and GFP was detected in the outer membrane of liver cells (Supplementary Fig. 1).

**Effects of liver HGF on hepatotropic metastasis.** We also evaluated the regulated liver HGF on GC liver metastasis and mouse survival. It is clearly shown that upregulated liver HGF promotes metastasis to the liver, whereas downregulation of liver HGF suppressed liver metastasis (Fig. 8a,b). High levels of liver HGF promote the growth of metastases, increasing the size and weight of metastatic focus (Fig. 8c,d). The expression of HGF and c-MET was also checked in GC liver metastases and para-carcinoma tissues, and HGF is highly expressed in liver tissues but not in the metastases (Fig. 8e,h). Phosphorylation of c-MET is also characterized using immunohistochemistry assay and western

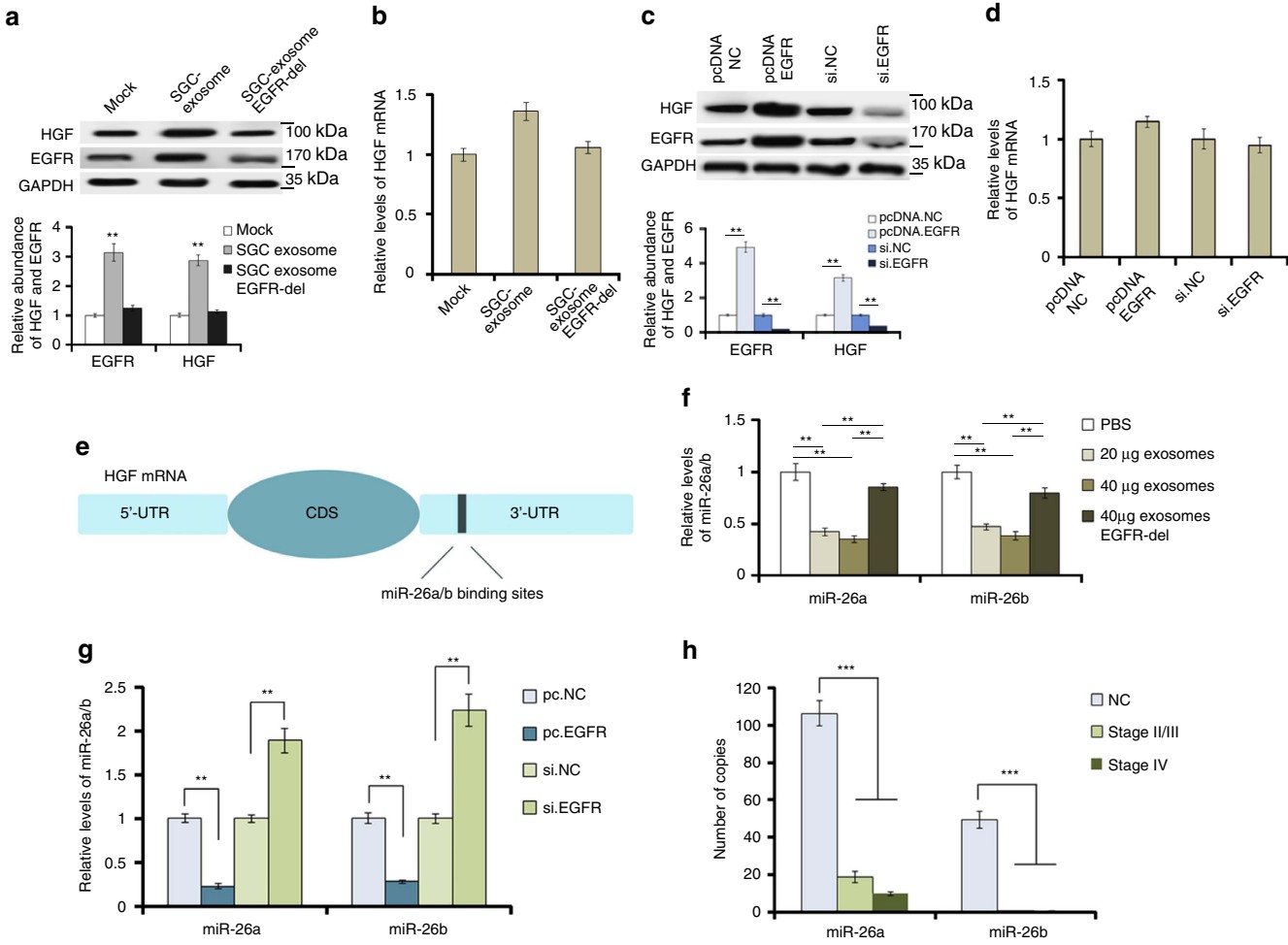

**Figure 4 | SGC-exosomes-mediated EGFR activates liver HGF by suppressing miR-26a/b expression.** Forty micrograms of exosomes were used to culture with $1 \times 10^6$ primary liver cells seeded in a six-well plate. (**a,b**) Effects of SGC exosomes delivered EGFR on HGF protein levels (**a**) and mRNA levels (**b**) in mixed primary liver cells ($n = 3$). (**c,d**) Effects of EGFR on the expression of HGF protein (**c**) and mRNA (**d**) ($n = 3$). (**e**) The binding sites of miR-26a/b in the 3′-UTR of HGF mRNA. (**f**) SGC-exosomes decrease liver miR-26a/b levels ($n = 3$). (**g**) EGFR is negatively related with miR-26a/b in liver cells ($n = 3$). (**h**) Absolute quantification of serum miR-26a/b in the progression of GC ($n = 150$). The data represent the mean ± s.e.m. **$P < 0.01$, ***$P < 0.001$ (Student's $t$-test).

blotting assay; as is expected, high levels of liver HGF promotes the activation of c-MET in metastases (Fig. 8f,g).

Therefore, liver HGF is closely linked with the process of liver metastasis in mouse model.

## Discussion

Liver is linked with the other gastrointestinal organs by hepatic portal vein, which is conducive to metastasis of GC, colorectal cancer and pancreatic cancer. However, this cannot fully explain liver metastasis of the other types of tumours, such as breast cancer, lung cancer and renal cancer[23]. As the Stephen Paget's hypothesis, studies have been focused on identifying cell-intrinsic determinants of organ-specific metastasis[24–27]. Recently, exosomes have been proved to play a key role in determining organ-specific metastasis[28–30]. Our study illustrated exosomes secreted from cancer cells can regulate the microenvironment of liver to prepare favourable conditions for future metastasis.

In recent years, exosomes have been a provocative topic in both the early detection of malignant tumours and signalling transduction between cells[31–33]. These cell-derived small particles are also used as a safe vehicle for the delivery of targeted drugs, as well as miRNAs and siRNAs[34–37]. Exosomes have been known to mediate the immune escape, drug resistance

and angiogenesis of tumours[7,38,39]. However, the role of exosomes in the communication between primary focus and future metastatic organs is little known.

Although EGFR has been reported to be delivered by tumour-derived exosomes[11,22], the role of secreted EGFR in the process of tumour metastasis remains unknown. We found that SGC-exosomes can transport EGFR into the liver and these EGFR is finally located in the membrane of stromal liver cells. Our study first revealed the location of tumour-secreted EGFR in target organs, explaining how membrane receptors can be transported between cells. Exosome-mediated EGFR activates liver HGF, thus preparing 'soil' for future cancer cell metastasis. This signalling pathway comprising exosome EGFR, liver miR-26a/b and HGF illustrates the novel mechanism involved in liver metastasis of GC.

In this study, EGFR in SGC cells and exosomes was knocked down by using siRNA and these EGFR-absent exosomes lost the function to promote liver HGF expression. However, the silencing of EGFR may lead to the change of content in exosomes; more convincing data can be provided by cetuximab or exosome blocking.

It is believed that exosomes derived from the other types of cancer can also regulate liver microenvironment to facilitate the

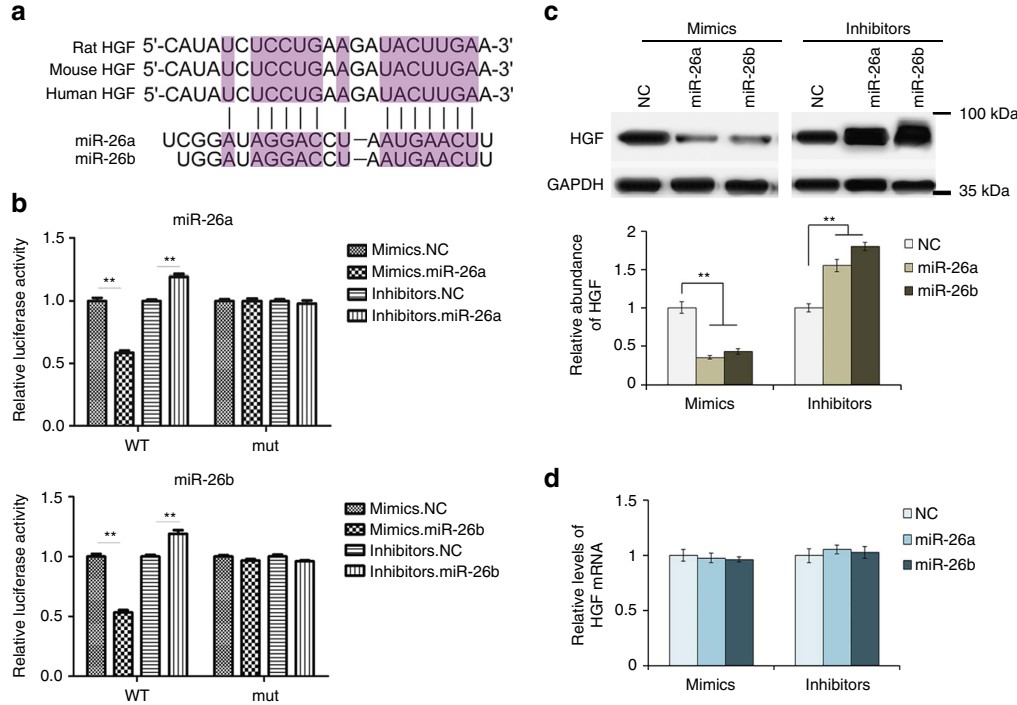

**Figure 5 | Identification of HGF as a direct target of miR-26a/b.** (**a**) Predicted binding sites of miR-26a/b within the 3′-UTR of HGF mRNA. (**b**) Direct recognition of HGF 3′-UTR by miR-26a and miR-26b. Primary liver cells were co-transfected with firefly luciferase reporters containing either wild-type or mutant (mut) HGF 3′-UTR with miR-26a/b mimics, inhibitors and the corresponding normal control. The relative luciferase levels were detected using a luciferase kit at 24–36 h after transfection ($n = 3$). (**c**) Western blot analysis of HGF expression in primary liver cells with the overexpression or suppression of miR-26a or miR-26b ($n = 3$). (**d**) Relative levels of HGF mRNA in liver cells transfected with miR-26a/b mimics or inhibitors ($n = 3$). The data represent the mean ± s.e.m. **$P < 0.01$ (Student's t-test).

formation and growth of liver metastases. Abundant HGF makes the liver one of the common metastatic sites for multiple tumours. Tumour-derived exosomes regulated the signalling pathways in the liver and makes it an ideal target for malignancy. The change of gene expression of liver cells by tumour-derived exosomes may lead to disorder, fibrosis and field cancerization in the liver, although this still needs more exploration.

Overall, our findings demonstrate an important role for tumour-derived exosomes in dictating liver-specific metastasis by remodelling liver microenvironment. Our results support the 'seed and soil' hypothesis, uncovering the novel mechanism of liver-tropism metastasis.

## Methods

**Human tissue.** Human GC liver metastatic focuses and paired adjacent non-cancerous tissues were derived from patients undergoing a surgical procedure at the Tianjin Medical University Cancer Institute and Hospital (Tianjin, China). Both tumour tissues and non-cancerous tissues were confirmed histologically. The pathological type of each cancer was determined to be glandular carcinoma. Written consent was provided by all of the patients and the Ethics Committee of Tianjin Medical University Cancer Institute and Hospital approved all aspects of this study. Tissue fragments were immediately frozen in liquid nitrogen at the time of surgery and stored at −80 °C.

**Animals.** Male nude mice (BALB/c-nu, 6~8 weeks) were housed in a pathogen-free animal facility with access to water and food, and allowed to eat and drink *ad libitum*. All of the experimental procedures were performed in accordance with protocols approved by the Institutional Animal Care and Research Advisory Committee of Nanjing University.

**Cell culture.** SGC7901 (human gastric adenocarcinoma cell) was bought from cell bank of Chinese Academy of Sciences (Shanghai, China) and was cultured in DMEM medium (Gibco, USA); SGC 7901 cells were tested for mycoplasma contamination before use; primary mouse liver cells were obtained from the livers of C57BL/6J mice (6–8 weeks of age) and were cultured in RPMI 1640 (Gibco);

both were supplemented with 10% fetal bovine serum (Gibco) in a humidified incubator at 37 °C with 5% $CO_2$.

**Isolation of exosomes from medium and serum.** Exosomes were isolated from cell culture medium by differential centrifugation, according to previous publications[40]. After removing cells and other debris by centrifugation at 300 g and 3,000 g, the supernatant was centrifuged at 10,000 g for 30 min to remove shedding vesicles and the other vesicles with bigger sizes. Finally, the supernatant was centrifuged at 110,000 g for 70 min (all steps were performed at 4 °C); exosomes were collected from the pellet and re-suspended in PBS. Sr-exosomes were isolated by using an exosome isolation kit (Thermo).

**Nanoparticle tracking analysis.** The number and size of exosomes were directly tracked using the Nanosight NS 300 system (NanoSight Technology, Malvern, UK)[41,42]. Exosomes were re-suspended in PBS at a concentration of 5 µg ml$^{-1}$ were further diluted 100- to 500-fold, to achieve between 20 and 100 objects per frame. Samples were manually injected into the sample chamber at ambient temperature. Each sample was configured with a 488 nm laser and a high-sensitivity scientific complementary metal-oxide semiconductor (sCMOS) camera, and was measured in triplicate at camera setting 13 with an acquisition time of 30 s and a detection threshold setting of 7. At least 200 completed tracks were analysed per video. Finally, data were analysed using the NTA analytical software (version 2.3).

**Transmission electron microscopy assay.** For conventional transmission electron microscopy, the exosome pellet was placed in a droplet of 2.5% glutaraldehyde in PBS buffer at pH 7.2 and fixed overnight at 4 °C. Samples were rinsed in PBS buffer (3 times, 10 min each) and post fixed in 1% osmium tetroxide for 60 min at room temperature (RT). The samples were then embedded in 10% gelatin and fixed in glutaraldehyde at 4 °C, and cut into several blocks (<1 mm³). The samples were dehydrated for 10 min each step in increasing concentrations of alcohol (30, 50, 70, 90, 95 and 100% × 3). Pure alcohol was then exchanged by propylene oxide and specimens were infiltrated with increasing concentrations (25, 50, 75 and 100%) of Quetol-812 epoxy resin mixed with propylene oxide for a minimum of 3 h per step. Samples were embedded in pure, fresh Quetol-812 epoxy resin and polymerized at 35 °C for 12 h, 45 °C for 12 h and 60 °C for 24 h. Ultrathin sections (100 nm) were cut using a Leica UC6 ultra-microtome and post stained with uranyl acetate for 10 min and with lead citrate for 5 min at RT before observation in a FEI Tecnai T20 transmission electron microscope, operated at 120 kV.

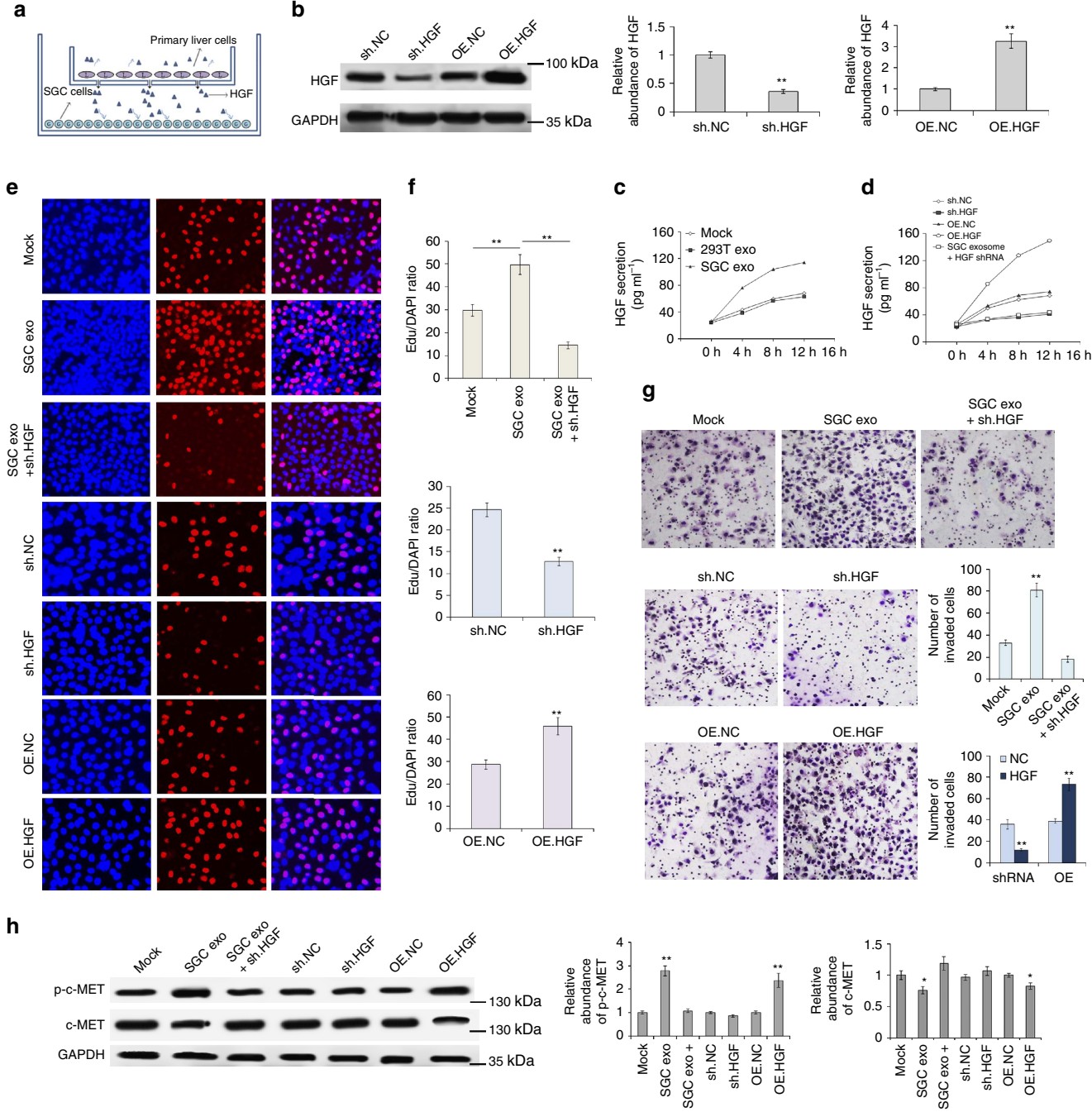

**Figure 6 | Upregulated liver paracrine HGF promotes proliferation and invasion of SGC7901 cells.** Pretreated mixed liver cells were co-cultured with SGC7901 cells using a 0.4 μm membrane and the biology behaviour of SGC7901 cells were measured subsequently. (**a**) Schematic representation of the *in vitro* model for cell co-culture. (**b**) Western blot analysis of HGF expression in primary liver cells treated with HGF shRNA and HGF-overexpressing lentivirus (OE.HGF) ($n = 3$). (**c**) Effects of SGC exosomes and control exosomes on the HGF secretion from mixed liver cells ($n = 3$). (**d**) ELISA analysis of HGF secretion from primary liver cells treated with HGF siRNA or OE.HGF lentivirus ($n = 3$). (**e–g**). Upregulated liver HGF promotes growth (**e**) and invasion (**g**) of SGC7901 cells ($n = 3$). (**h**) c-MET and p-c-MET expression in mixed liver cells treated as above ($n = 3$). The data represent the mean ± s.e.m. $*P < 0.05$, $**P < 0.01$ (Student's *t*-test).

**Immunofluorescence.** Cells were cultured on four-well chamber slides. At the time of harvest, cells were fixed with 4% paraformaldehyde and then permeabilized with 0.01% Triton X-100 for 10 min. Then cells were treated with anti-desmin antibody (Immunoway, YT1326), anti-α-SMA antibody (1:50; Santa Cruz; sc-53142), and anti-F4/80 antibody (1: 100; Abcam; ab100790). In addition, all samples were treated with 4′,6-diamidino-2-phenylindole dye for nuclear staining (358 nm). For confocal microscopy, a Nikon C2 Plus confocal microscope was used.

**RNA isolation and quantitative reverse transcriptase–PCR.** Assays to quantify mature miRNAs were conducted as previously described, with slight modifications[43,44]. Total RNA was extracted from the cultured cells and tissues using TRIzol Reagent (Invitrogen) according to the manufacturer's instructions. miRNA determination was performed using Taqman miRNA probes (Applied Biosystems, Foster City, CA). All of the reactions were run in triplicate. After the reactions were complete, the cycle threshold ($C_T$) data were determined using fixed threshold settings and the mean $C_T$ was determined from triplicate PCRs. A comparative $C_T$

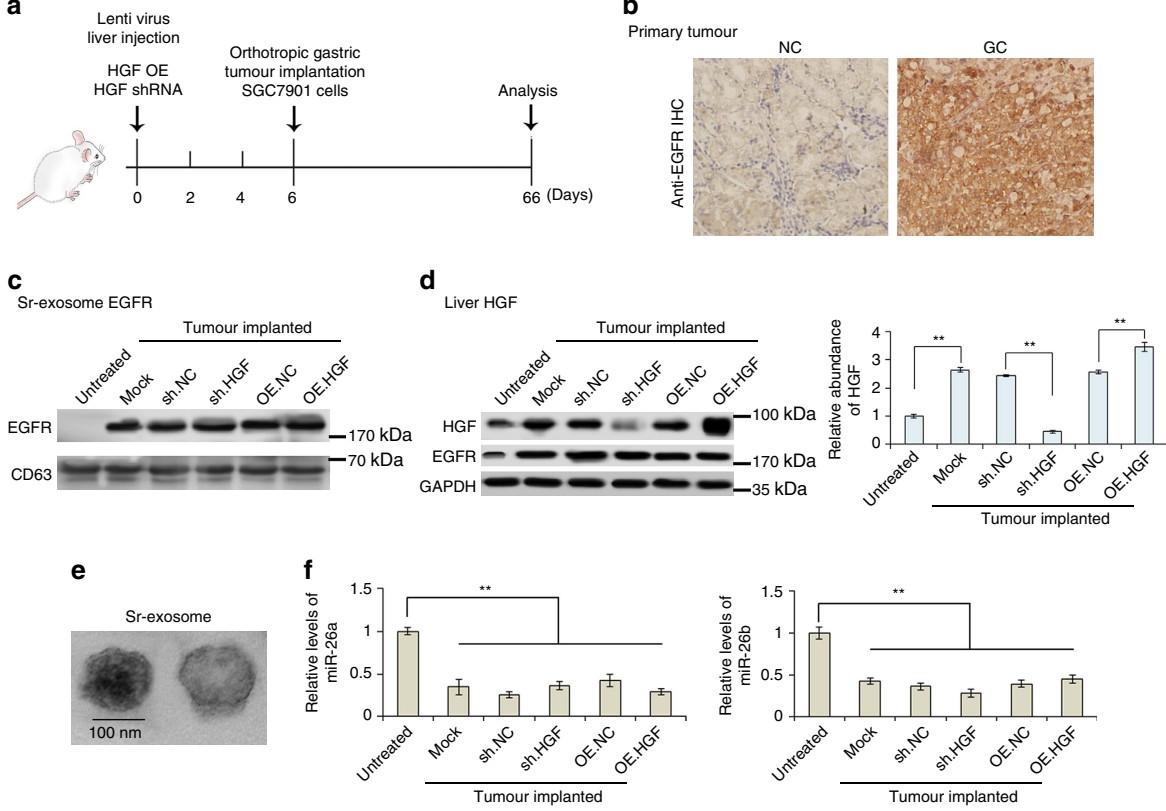

**Figure 7 | *In vivo* verification for exosome-EGFR activates liver HGF by inhibiting miR-26a/b.** The livers of mice were pre-treated with HGF shRNA or HGF overexpressing lentivirus by multi-point injection, followed by orthotopic tumour implantation on the 6th day; finally, mice were killed and data were analysed on the 66th day. (**a**) A flow chart depicting the *in vivo* experimental design. (**b**) Immunohistochemistry (IHC) analysis of EGFR expression in primary gastric tumour. (**c**) Western blotting analysis of exosome EGFR in the serum of tumour-implanted mice ($n = 30$). (**d**) Levels of HGF in liver tissues of GC tumour-implanted mice ($n = 30$). (**e**) Electron-microscope scanning of exosomes isolated from mouse serum. (**f**) Relative levels of miR-26a/b in mouse liver tissues ($n = 30$). The data represent the mean ± s.e.m. **$P < 0.01$ (Student's *t*-test).

method was used to compare each condition to the control reactions. U6 small nuclear RNA was used as an internal control of miRNAs and the mRNA levels were normalized to glyceraldehyde 3-phosphate dehydrogenase (GAPDH). The relative amount of gene normalized to control was calculated with the equation $2^{-\Delta CT}$, in which $\Delta C_T = C_T \text{ gene} - C_T \text{ control}$.

Primers of HGF and GAPDH were as follows: 5′-AGAAGGCTGGGGCT CATTTG-3′ (GAPDH, sense), 5′-AGGGGCCATCCACAGTCTTC-3′ (GAPDH, anti-sense); and 5′-CCTGGTGCTACACGGGAAAT-3′ (HGF, sense), 5′-CACAT CCACGACCAGGAACA-3′ (HGF, anti-sense).

**The miRNA target prediction.** The miRNA target prediction and analysis were performed with the algorithms from TargetScan (http://www.targetscan.org/), PicTar (http://pictar.mdc-berlin.de/) and miRanda (http://www.microrna.org/).

**Luciferase assay.** The reporter plasmid p-MIR-HGF containing the predicted miR-26 targeting regions was designed by Genescript (Nanjing, China). Part of the wild-type and mutated 3′-UTR of HGF was cloned immediately downstream of the firefly luciferase reporter. The 2 mg of β-galactosidase expression vector (Ambion) was used as a transfection control. For the subsequent luciferase reporter assays, 2 mg of firefly luciferase reporter plasmid, 2 mg of β-galactosidase vector and equal doses (200 pmol) of mimics, inhibitors or scrambled negative control RNA were transfected into the prepared cells. At 24 h after transfection, cells were analysed using the Dual Luciferase Assay Kit (Promega) according to the manufacturer's instructions. Each sample was prepared in triplicate and the entire experiment was repeated three times.

**Cell proliferation assay.** SGC7901 cells were incubated in 50 μM Edu (RiboBio Inc.) for 6 h and fixed with 4% paraformaldehyde for 30 min at 25 °C. Next, the cells were washed in PBS ($2 \times 5$ min, RT) and then permeabilized using PBS containing 0.3% Triton X-100 for 10 min. After extensive washes in PBS, the cells were incubated in Apollo staining solution (RiboBio, Inc.) for 20 min, washed with

NaCl/Pi ($3 \times 10$ min, RT) and then incubated in 4′,6-diamidino-2-phenylindole (1:2,500; Roche Diagnostics, Mannheim, Germany) for 10 min at RT.

**Cell migration assay.** The migration ability of SGC7901 cells was tested in a Transwell Boyden Chamber (6.5 mm, Costar) with polycarbonate membranes (8 μm pore size) on the bottom of the upper compartment. Cells were suspended in serum-free DMEM medium at a total amount of $1 \times 10^5$ cells; simultaneously, 0.5 ml DMEM with 10% fetal bovine serum was added to the lower compartment and the Transwell-containing plates were incubated for 6–8 h. At the end of the incubation, cells that have entered the lower surface of the filter membrane were fixed with 90% ethanol for 15 min at RT and stained with 0.1% crystal violet solution. Images of migrant were captured by photo-microscope and cell migration was quantified by blind counting with three fields per chamber.

**ELISA assay.** HGF secretion was determined using an ELISA kit according to the manufacturer's instructions (Sigma, RAB0212).

**Western blotting.** The HGF, cMET and EGFR expression was assessed by western blotting analysis and samples were normalized to GAPDH. Protein extraction was blocked with PBS-5% fat-free dried milk at RT for 1 h and incubated at 4 °C overnight with anti-HGF (1:1,000, Abcam), anti-EGFR (1:5,000, Abcam), anti-cMET (1:1,000, Abcam), anti-p-cMET (1:1,000, Abcam), anti-CD63 (1:2,000, Abcam), anti-TSG101 (1:1,000, Santa Cruz), anti-Alix (1: 1,000, Santa Cruz), anti-F4/80 (1:1,000, Abcam), anti-desmin (1:1,000, Immunoway), anti-α-SMA (1:1,000, Santa Cruz) and anti-GAPDH (1:3,000, Santa Cruz) antibodies, respectively.

**Immunohistochemistry.** The tumours were fixed in 4% paraformaldehyde, embedded in paraffin, sectioned and then stained with anti-c-MET antibodies (Abcam), anti-p-c-MET antibodies (Abcam) and anti-HGF antibodies (Abcam). Quantitative analysis was conducted by quantifying the fluorescence intensity from at least five sections.

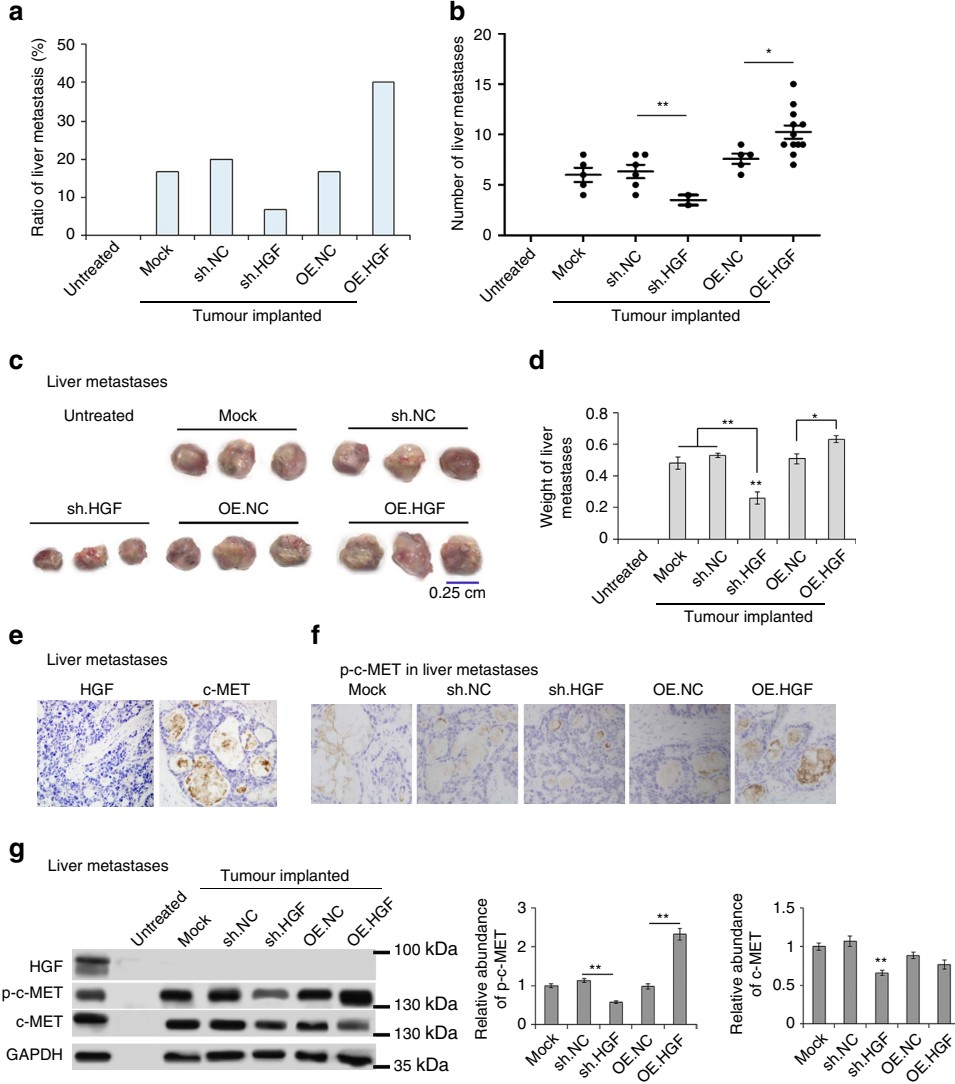

**Figure 8 | Effects of upregulated liver HGF on hepatotropic metastasis.** (**a**) Ratio of liver metastasis in GC tumor-implanted mice ($n = 30$). (**b**) Number of metastases in the liver of each mouse (the number in each group was marked). (**c**) Images of liver metastases in mice ($n = 30$). (**d**) The weight of tumours in **c**. (**e**) Immunohistochemistry (IHC) analysis of HGF and c-MET expression in GC liver metastases. (**f**) IHC analysis of p-c-MET in the liver metastases of each group. (**g**) Western blot analysis of HGF, p-c-MET and c-MET in GC liver metastases. The data represent the mean ± s.e.m. *$P < 0.05$, **$P < 0.01$ (Student's t-test).

**Establishment of tumour in nude mice.** SGC7901 cells were injected into nude mice by orthotopic implantation[45]. Briefly, $1 \times 10^7$ cells were first injected subcutaneously for one mouse and tumours were removed and divided into small pieces on the 15th day, each with 0.3 g, and the divided small tumours were implanted into the gastric subserosal haematoma.

**Statistical analyses.** All data were representative of five or six independent experiments. Data were expressed as mean ± s.e. of at least five separate experiments. Statistical significance was considered at $P < 0.05$ using the Student's t-test. In this study, *$P < 0.05$, **$P < 0.01$ and ***$P < 0.001$.

**Data availability.** All the data are available within the Article and Supplementary Information file, or available from the authors upon request.

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

## Acknowledgements

This work was supported by grants from the National Natural Science Foundation of China (numbers 81602158, 81372394, 81602156 and 81572321) and Tianjin Health and Family Planning Commission Foundation of Science and Technology (15KG142). This work was also supported by Tianjin Science Foundation (number 15JCYBJC28200), CSCO-Merck Serono Oncology Research Fund (Y-MX2015-092) and Doctoral foundation of Tianjin Medical University Cancer Institute and Hospital (B1502). The funders had no role in study design, collection, analysis and interpretation of data, in the writing of the report and in the decision to submit this article for publication.

## Author contributions

H.Z., T.D. and R.L. performed most of the experiments, analysed data and wrote the manuscript. M.B., L.Z., X.W. and S.L. reviewed and edited the manuscript. X.W., H.Y., J.L., T.N., D.H., H.L. and L.Z. performed some experiments. Y.B. and G.Y. designed the experiments and edited the manuscript. Y.B. is the guarantor of this work and had full access to all of the data in the study, and takes responsibility for the integrity of the data and the accuracy of the data analysis.

## Additional information

**Competing interests:** The authors declare no competing financial interests.

**Publisher's note**: 

