## [Peer Review File · Nature Communications]

Reviewers' comments:

Reviewer #1 (Remarks to the Author):

Summary:

The authors report on increased levels of circulating EGFR-bearing exosomes in gastric cancer patients that correlated with disease stage and was concomitant with increased liver HGF levels and increased c-Met expression in GC liver metastases. Using a murine gastric cancer cell line (SGC7901) they show that exosomes secreted by these cells contained EGFR and that upon co-culture, the EGFR was internalized by liver cells resulting in increased HGF production by the recipient cells. They show furthermore, that the increase in HGF levels was due to suppression of miR-26a/b levels and that the levels of these miRNAs were reduced in livers of GC patients. Finally, they show that in mice implanted with SGC7901 cells, pre-treatment with HGF shRNA reduced the number of liver metastases whereas pre-treatment with HGF overexpressing lentiviruses enhanced liver metastasis.

The data presented in this manuscript are interesting and there are some novel elements, such as the role of exosomal EGFR in miR-26a/b upregulation in the liver and its potential contribution to increased HGF production and consequently, liver metastasis. However, as detailed below, some critical experimental data are lacking and required to resolve outstanding issues.

General comments

1. The authors used primary liver cultures for many of the experiments. No characterization of these cells is provided. Are these primary hepatocyte cultures or mixed liver cell types? In the liver, HGF is produced mainly by stromal cells such as Kupffer, stellate and sinusoidal endothelial cells and not by hepatocytes (e.g. Fajardo-Puerta AB, et al. J Clin Pathol 2016;0:1-5), so the high expression seen in primary cultures (for example as shown in Fig 4C) is surprising, if these indeed are primary hepatocyte cultures. The authors should characterize these cells, provide information on the liver cell types in their cultures and if hepatocytes, address the atypical high expression levels seen in untreated cultures.

2. The authors speculate that HGF upregulation due to miRNA suppression leads to c-Met activation and signaling in metastatic tumor cells. However no evidence to directly support c-Met activation is provided. The data shows c-Met overexpression in GC cells and in liver metastases. To support their conclusion, the authors will need to demonstrate c-Met activation (e.g. phosphorylation of c-Met or downstream substrates) in these cells.

Specific comments

Fig 2E- Based on this image alone, it is not possible to categorically state that EGFR-GFP is located in the membranes of recipient cells. This can be demonstrated through subcellular fractionation and/or co-immunostaining with a membrane marker. Higher power images should be provided.

Fig 5. In this figure, the authors show that co-culture of GC cells with primary cultures of liver cells overexpressing HGF or pretreated with GC exosomes increased growth and invasion of the cancer cells. To support their conclusion, the authors should have shown that administering HGF shRNA to SGC7901 exosome-treated liver cells reduced their ability to stimulate the growth and invasion of the cancer cells. This critical experiment is missing.

Fig 6. The experimental design is not well justified. It is not clear why HGF shRNA or lentiviruses were injected 6 days prior to tumor implantation rather than post implantation when HGF levels are presumably upregulated. Also not clear how shRNA or viruses were injected and where in the liver they were taken up. The authors should demonstrate uptake by specific cell types in the liver (e.g. Kupffer cells, hepatocytes, etc.) identify the HGF producing cell and show evidence that these cells are indeed targeted by the shRNA or lentiviruses.

Fig 6D. The increase in liver HGF in tumor implanted vs non-implanted mice is not very

compelling. Also, not clear which cells are producing it.

Fig 7E. As indicated above, the relevant parameter is c-Met activation levels. The authors should also provide IHC evidence for increased phospho-c-Met levels.

Minor comments

1. Institutional approval protocol number for the use of human specimens should be provided.
2. The authors describe tumor implantation method as "in situ planting of gastric cancer". Not clear what is meant. Is this an orthotopic implantation model? Reference cited (# 24) is irrelevant, as it deals with liver cancer.
3. Parenthesis in line 194 should read Figure 1D and E.
4. The manuscript can benefit from language editing and spell check.

Reviewer #2 (Remarks to the Author):

In the present manuscript the authors investigate the mechanisms of the metastatic process. They propose that gastric cancer cells produce exosomes containing EGFR and that the transfer of exosomal EGFR to the recipient liver cells promotes development of liver metastases. The key event in this hypothetical model of the inter-cellular signaling is an increased production of HGF in the liver, which is mediated by the exosome-dependent suppression of two miRNAs. The importance of the increased level of HGF for liver metastasis is well documented in in vitro and in vivo experiments. That extracellular vesicles (EV) produced by gastric cancer cells are capable of potentiating HGF expression and the metastatic process is also convincingly demonstrated. However, a novel aspect of the process, an involvement of exosome-transferred EGFR, is not supported by the experimental evidence but is rather based on correlative observations (see major comments below). The major concern is that the method of exosome purification used in this study does not allow separation of "true" exosomes (originated from multi-vesicular endosomes) from other EVs. Additional step of sucrose or Optiprep density gradient centrifugation is necessary to purify exosomes. Furthermore, Fig. 1 illustrates the presence of CD63 but does not show an enrichment of this and other exosomal markers. (Please see Kowal et al., 2016 PNAS for an example of the rigorous purification and classification of various EVs and their markers.). Another major concern is that the amount of EGFR in exosomes is not compared with the concentration of EGFR in the donor cancer and recipient liver cells. Most liver cells express relatively high levels of EGFR. It is unclear if transport of additional EGFR by EVs can significantly increase total EGFR levels in these cells, which would lead to a qualitatively different signaling outcome.

Other major comments

- 1) Fig. 2B. What are the GES-1 cells? Do they express EGFR levels comparable to that in SGC cells? If not, comparison of the amounts of EGFR in EVs is not helpful.
- 2) Fig. 2E. The pattern of EGFR-GFP localization in the recipient cell does not look like the plasma membrane. The cell surface localization of transferred EGFR is one of the key points of the proposed model and should be directly shown. Can the EV transfer of endogenous EGFR, not overexpressed EGFR-GFP, be demonstrated?
- 3) Fig. 2D. The effect is apparent when a concentrated preparation of purified EV is used. Can the transfer of the EV material be demonstrated in the co-culture of donor and recipient cells?
- 4) Fig. 3 demonstrates the effect of EV on HGF expression level but does not show that this effect is EGFR dependent. Multiple components in EVs, including miRNAs, may mediate observed effects.
- 5) Fig. 5C. Does the concentration of EVs used in this experiment resemble the EV concentration in the serum?
- 6) Fig. 6. Evidence that EV-containing EGFR is responsible for the observed effect is lacking.

Minor comments

- 1) Fig. 7B. Is there statistical significance of the increase in the OE-HGF variant?
- 2) Fig. 7D. It is unclear why HGF overexpression does not increase liver metastases.
- 3) Fig. 7F. The lack of HGF in metastatic nodes seems counterintuitive since HGF is necessary for their formation. This issue deserves explanation and discussion.

Reviewer #3 (Remarks to the Author):

The investigators study the role of EGFR containing exosomes in promoting gastric cancer metastasis. In its present form, the study is not complete and disjointed. Some of the claims in the article are based on assumptions with no data to support it. Though the manuscript is easy to follow in most places, the text needs significant improvement. Lot of spelling mistakes in text, figures and figure legends. The manuscript needs additional data to support their big claims and hence not convincing in its present form. Some of the claims are:

- 1, EGFR is transferred via exosomes secreted by GC cells to liver cell membrane (no convincing data provided)
- 2, Exosomes increase metastasis and HGF secretion in vivo (need more data)

Some of the critical issues are mentioned below.

Major points

1. The article is not clear and confusing with the aspect of EGFR, HGF and exosomes. The authors need to understand that exosomes contain thousands of proteins and RNA. Hence, to claim the functional effect to EGFR and HGF alone is not justified given the experimental setup used in the study.
2. More work is needed to claim EGFR as the single entity in this process. Can the authors block EGFR in exosomes using cetuximab or other inhibitors in vivo? What will happen to liver HGF and metastasis then? Can the authors provide this data?
3. Additionally, what happens if the amount of secreted exosomes from gastric cancer cells is decreased? This is standard in the field of exosomes (PMID: 22635005). Can the authors block the release of exosomes from GC cells in vivo using inhibitors or gene knockdown/knockout? What will happen to liver HGF and metastasis then? Can the authors provide this data?
4. Authors neglect some of the previous work done on metastasis and organotropism in the context of exosomes (PMID: 26524530, 25985394, 22635005). The articles are published from 2012 and hence could have been included.
5. The exosomes characterization is poor. Can the authors provide Western data for exosomes markers Alix and Tsg101 as referred to as minimal standards publication (PMID: 25536934)? Both for serum and cell exosomes.
6. CD63 banding pattern is odd. Can the authors add where the antibody was purchased from and what dilution was used?
7. What is the concentration of exosomes used in Figure 2? Can the authors add this in text, figure and figure legend? (??? ug/mL) Also, how many cells were used for primary liver? These data are important to reproduce the results. Similarly, for all other places such as 3A etc.
8. Fig 2D seems to be not correct and not done well. What is the negative control for the dye alone? What if the dye is diffusing by itself? How can the dye stain all (area as well as all) of the cell with very little exosomes? If more exosomes are used, is the concentration physiologically relevant? Never seen this for last 10 years. My observation, dye is diffusing.
9. Can the authors show Fig 2E as similar to Fig 2D with DAPI. Also, more cells - not just one cell.
10. Can the authors show that EGFR is overexpressed in liver cells? There is no data on this in Fig 6 when the authors claim the whole process is driven by EGFR. As the authors have EGFR-GFP GC cells, can't they do the xenograft on these to show EGFR is in the membrane of liver tissue in vivo (as they claim).
11. Fig 3F - 20ug exosomes. Need to add per mL or X no of cells?

12. Authors say exosomes fuse with liver cell membrane? How do they know it is fusion? What data is provided? This is complete speculation.
13. Can the patient numbers be written in the results text for ease of reading? Also, mention in figure legend that the figure is representative (where applicable).
14. What are the inhibitors of miR26a/b? What are the mimics? It is not clear.
15. Why is mRNA of HGF not decreasing in figure 4D? The authors also say, as expected. If miR26a/b degrades RNA, would you not expect to see reduction in RNA levels as well?
16. As the authors used 0.4 μm pore, all exosomes ($<0.1 \mu\text{m}$) and secreted factors can pass through. How can the authors attribute the function to HGF alone? Can the authors block HGF by antibodies? Or block the receptors in the recipient cells? In this form, there are too many factors and hence HGF alone is not convincing.
17. Is the results similar to other GC cell models as well?

Minor points

1. Second paragraph in introduction starts with "Extracellular vesicles".
2. Figure legend says "Photos" of exosomes. This is electron micrograph. Please see other exosome articles on how these are referred previously.
3. Results (line 194), Figure 2D should be 1D.
4. Figure 1B, spelling mistake on exosomes

Response to referees' comment:

Reviewer #1

General comments

1. The authors used primary liver cultures for many of the experiments. No characterization of these cells is provided. Are these primary hepatocyte cultures or mixed liver cell types? In the liver, HGF is produced mainly by stromal cells such as Kupffer, stellate and sinusoidal endothelial cells and not by hepatocytes (e.g. Fajardo-Puerta AB, et al. J Clin Pathol 2016;0:1-5), so the high expression seen in primary cultures (for example as shown in Fig 4C) is surprising, if these indeed are primary hepatocyte cultures. The authors should characterize these cells, provide information on the liver cell types in their cultures and if hepatocytes, address the atypical high expression levels seen in untreated cultures.

Answer: We do know that HGF is expressed in stromal cells; in this article, primary liver cells contain mix liver cell types, and this is why HGF is detected in mixed primary liver cells. We characterized these cell types using several markers, such as CD166, CD31, these results were added in supplemental data.

2. The authors speculate that HGF upregulation due to miRNA suppression leads to c-Met activation and signaling in metastatic tumor cells. However no evidence to directly support c-Met activation is provided. The data shows c-Met overexpression in GC cells and in liver metastases. To support their conclusion, the authors will need to demonstrate c-Met activation (e.g. phosphorylation of c-Met or downstream substrates) in these cells.

Answer: The expression of phosphorylated c-MET was determined by western blot; it was showed that up-regulated HGF may promote the activation of c-MET. These data were added in Figure 1B, Figure 5H and Figure 7F.

Specific comments

1. Fig 2E- Based on this image alone, it is not possible to categorically state that EGFR-GFP is located in the membranes of recipient cells. This can be demonstrated through subcellular fractionation and/or co-immunostaining with a membrane marker. Higher power images should be provided.

Answer: Exosomes containing GFP-tagged EGFR were used to treat primary liver cells; and E-Cadherin was used as the membrane marker. Results from the immunofluorescence showed that EGFR-GFP was co-localized with E-cadherin (Figure 2E).

2. Fig 5. In this figure, the authors show that co-culture of GC cells with primary cultures of liver cells overexpressing HGF or pretreated with GC exosomes increased growth and invasion of the cancer cells. To support their conclusion, the authors should have shown that administering HGF shRNA to SGC7901 exosome-treated liver cells reduced their ability to stimulate the growth and invasion of the cancer cells. This critical experiment is missing.

Answer: According to the reviewer's advice, we used SGC exosomes and HGF shRNA to treat primary liver cells simultaneously; and the secretion of liver HGF was determined (Figure 5D), as well as the cell growth and invasion of cancer cells (Figure 5E-G).

3. Fig 6. The experimental design is not well justified. It is not clear why HGF shRNA or lentiviruses were injected 6 days prior to tumor implantation rather than post implantation when HGF levels are presumably upregulated. Also not clear how shRNA or viruses were injected and where in the liver they were taken up. The authors should demonstrate uptake by specific cell types in the liver (e.g. Kupffer cells, hepatocytes, etc.) identify the HGF producing cell and show evidence that these cells are indeed targeted by the shRNA or lentiviruses.

Answer: The lenti-viruses containing HGF shRNA or HGF-overexpressing sequence were injected into mouse liver by in situ multi-point injection. These lenti-viruses were injected 6 days prior to tumor implantation because it takes several days for lenti-viruses to up-regulate or down-regulate a specific gene in vivo.

The in vivo experiments are designed to demonstrate that HGF levels in liver microenvironment play a key role in promoting hepatotropic metastasis. Lenti-viruses are known to infect cells without specificity, but they can truly regulate liver HGF.

4. The increase in liver HGF in tumor implanted vs non-implanted mice is not very compelling. Also, not clear which cells are producing it.

Answer: We re-detected liver HGF in mice, and western blot analysis showed that liver HGF was up-regulated 2.5 folds in the group of tumor implanted.

5. Fig 7E. As indicated above, the relevant parameter is c-Met activation levels. The authors should also provide IHC evidence for increased phospho-c-Met levels.

Answer: The expression of p-c-MET in liver metastases was determined by western blot, the data was added in Figure 7H.

Minor comments

1. The authors describe tumor implantation method as "in situ planting of gastric cancer". Not clear what is meant. Is this an orthotopic implantation model? Reference cited (# 24) is irrelevant, as it deals with liver cancer.

Answer: "in situ planting of gastric cancer" is the orthotopic implantation model, this was correct in the manuscript and Figures.

2. Parenthesis in line 194 should read Figure 1D and E.

Answer: This was corrected in the manuscript.

3. The manuscript can benefit from language editing and spell check.

Answer: We re-edited the language and checked the spelling in the manuscript.

Reviewer #2

Major concerns:

- a) The major concern is that the method of exosome purification used in this study does not allow separation of "true" exosomes (originated from multi-vesicular endosomes) from other EVs. Additional step of sucrose or Optiprep density gradient centrifugation is necessary to purify exosomes.

Answer: The size of exosomes was determined using a Nanoparticle Tracking Analysis; as is shown in Figure 2F, most of the exosomes were around 100 nm, which is the typical size for exosome. Therefore, we believe that they are true exosomes.

- b) Furthermore, Fig. 1 illustrates the presence of CD63 but does not show an enrichment of this and other exosomal markers. (Please see Kowal et al., 2016 PNAS for an example of the rigorous purification and classification of various EVs and their markers.).

Answer: Novel markers for exosomes, TSG101, Alix as well as CD63 were used as the markers of exosomes (Figure 1B, Figure 2B and Figure 6C). TSG101 and Alix showed an enrichment in exosomes compared with cells. (Figure2B)

- c) Another major concern is that the amount of EGFR in exosomes is not compared with the concentration of EGFR in the donor cancer and recipient liver cells. Most liver cells express relatively high levels of EGFR. It is unclear if transport of additional EGFR by EVs can significantly increase total EGFR levels in these cells, which would lead to a qualitatively different signaling outcome.

Answer: We checked the effects of SGC exosomes on the expression of liver EGFR (Figure 3A). It was showed that EGFR was significantly up-regulated by exosomes from cancer cells.

Other Major comments:

- 1) Fig. 2B. What are the GES-1 cells? Do they express EGFR levels comparable to that in SGC cells? If not, comparison of the amounts of EGFR in EVs is not helpful

Answer: GES-1 cells are normal gastric cells. This result was deleted from Figures.

- 2) Fig. 2E. The pattern of EGFR-GFP localization in the recipient cell does not look like the plasma membrane. The cell surface localization of transferred EGFR is one of the key points of the proposed model and should be directly shown. Can the EV transfer of endogenous EGFR, not overexpressed EGFR-GFP, be demonstrated?

Answer: Exosomes containing GFP-tagged EGFR were used to treat primary liver cells; and E-Cadherin was used as the membrane marker. Results from the immunofluorescence showed that EGFR-GFP was co-localized with E-cadherin (Figure 2E).

- 3) Fig. 2D. The effect is apparent when a concentrated preparation of purified EV is used. Can the transfer of the EV material be demonstrated in the co-culture of donor and recipient cells?

Answer: No. The exosomes need be labeled by membrane dye; and this method was widely used in recent studies.

- 4) Fig. 3 demonstrates the effect of EV on HGF expression level but does not show that this effect is EGFR dependent. Multiple components in EVs, including miRNAs, may mediate observed effects.

Answer: Levels of EGFR in SGC exosomes were decreased by transfection of EGFR siRNA into SGC7901 cells (Figure 2B). And these EGFR-absent exosomes lost the function of promoting HGF expression in primary liver cells (Figure 3A).

- 5) Fig. 5C. Does the concentration of EVs used in this experiment resemble the EV concentration in the serum?

Answer: The concentration used in this experiment was different from the exosome concentration in the serum. Serum exosomes are secreted from all tissues and all types of cell in the body.

- 6) Fig. 6. Evidence that EV-containing EGFR is responsible for the observed effect is lacking.

Answer:

Minor comments

- 1) Fig. 7B. Is there statistical significance of the increase in the OE-HGF variant?

Answer: Yes, $p < 0.05$.

- 2) Fig. 7D. It is unclear why HGF overexpression does not increase liver metastases.

Answer: We feel really sorry that two groups, 'OE.NC' and 'OE.HGF', were mixed mistakenly. In the corrected Figure 7C and 7D, the liver metastases in the OE.HGF group showed bigger size and weight compared with the control group.

- 3) Fig. 7F. The lack of HGF in metastatic nodes seems counterintuitive since HGF is necessary for their formation. This issue deserves exploration and discussion.

Answer: This does need more exploration in future study.

Reviewer #3

Major points

1. The article is not clear and confusing with the aspect of EGFR, HGF and exosomes. The authors need to understand that exosomes contain thousands of proteins and RNA. Hence, to claim the functional effect to EGFR and HGF alone is not justified given the experimental setup used in the study.

Answer: Levels of EGFR in SGC exosomes were decreased by transfection of EGFR siRNA into SGC7901 cells (Figure 2B). And these EGFR-absent exosomes lost the function of promoting HGF expression in primary liver cells (Figure 3A).

2. More work is needed to claim EGFR as the single entity in this process. Can the authors block EGFR in exosomes using cetuximab or other inhibitors in vivo? What will happen to liver HGF and metastasis then? Can the authors provide this data?

Answer: It is quite difficult for us to complete these in vivo experiments within three months. Moreover, cetuximab and the other inhibitors are believed to suppress EGFR function in exosomes and all types of cells, thus this experiment cannot demonstrate the function of exosome delivered EGFR. But we have provided data to show that EGFR-absent exosomes are not able to boost liver HGF expression (Figure 3A).

3. Additionally, what happens if the amount of secreted exosomes from gastric cancer cells is decreased? This is standard in the field of exosomes (PMID: 22635005). Can the authors block the release of exosomes from GC cells in vivo using inhibitors or gene knockdown/knockout? What will happen to liver HGF and metastasis then? Can the authors provide this data?

Answer: We cannot perform this experiment so far, because it remains known which gene determines exosome release from GC cells. But SGC exosome containing EGFR-GFP were injected into mice via tail vein, and we detected the EGFR-GFP in liver (Figure S2).

4. Authors neglect some of the previous work done on metastasis and organotropism in the context of exosomes (PMID: 26524530, 25985394, 22635005). The articles are published from 2012 and hence could have been included.

Answer: These articles were cited in the part of discussion (ref: 34-36).

5. The exosomes characterization is poor. Can the authors provide Western data for exosomes markers Alix and Tsg101 as referred to as minimal standards publication (PMID: 25536934)? Both for serum and cell exosomes.

Answer: Novel markers for exosomes, TSG101, Alix as well as CD63 were used as the

markers of exosomes (Figure 1B, Figure 2B and Figure 6C). TSG101 and Alix showed an enrichment in exosomes compared with cells. (Figure2B).

6. CD63 banding pattern is odd. Can the authors add where the antibody was purchased from and what dilution was used?

Answer: We re-performed western blot using CD63 antibody purchased from Abcam.

7. What is the concentration of exosomes used in Figure 2? Can the authors add this in text, figure and figure legend? (??? ug/mL) Also, how many cells were used for primary liver? These data are important to reproduce the results. Similarly, for all other places such as 3A etc.

Answer: The concentration of exosomes and number of liver cell were added in Figures and Figure legends.

8. Fig 2D seems to be not correct and not done well. What is the negative control for the dye alone? What if the dye is diffusing by itself? How can the dye stain all (area as well as all) of the cell with very little exosomes? If more exosomes are used, is the concentration physiologically relevant? Never seen this for last 10 years. My observation, dye is diffusing.

Answer: We performed this experiments again, and the new data was shown in Figure 2D. The PKH26-labeled exosomes entered into the cells without diffusing.

9. Can the authors show Fig 2E as similar to Fig 2D with DAPI. Also, more cells - not just one cell.

Answer: Exosomes containing GFP-tagged EGFR were used to treat primary liver cells; and E-Cadherin was used as the membrane marker. Results from the immunofluorescence showed that EGFR-GFP was co-localized with E-cadherin (Figure 2E).

10. Can the authors show that EGFR is overexpressed in liver cells? There is no data on this in Fig 6 when the authors claim the whole process is driven by EGFR. As the authors have EGFR-GFP GC cells, can't they do the xenograft on these to show EGFR is in the membrane of liver tissue in vivo (as they claim).

Answer: Liver-EGFR was checked using western blot analysis (Figure 6D). SGC exosome containing EGFR-GFP were injected into mice via tail vein, and we detected the EGFR-GFP in liver(Figure S2) .

11. Fig 3F - 20ug exosomes. Need to add per mL or X no of cells?

Answer: In Figure 3F, 20 or 40 μg exosomes were cultured with 1×10^6 primary liver cells. This was added in Figure legend and Results text.

12. Authors say exosomes fuse with liver cell membrane? How do they know it is fusion? What data is provided? This is complete speculation.

Answer: Exosomes can fuse with the recipient cells; this is the way exosomes play its role. But we still deleted 'fuse' in the text.

13. Can the patient numbers be written in the results text for ease of reading? Also, mention in figure legend that the figure is representative (where applicable)

Answer: Yes. The numbers of patients were added in the results text and figure legends.

14. What are the inhibitors of miR26a/b? What are the mimics? It is not clear.

Answer: mimics are chemosynthetic double-strand RNAs that can lead to up-regulation of a mature miRNA; while inhibitors are single-strand RNAs that can bind with a endogenic miRNA, resulting in the silencing of miRNA function. miRNA mimics and inhibitors are widely used in the area of miRNA research, thus it is not clearly described in text.

15. Why is mRNA of HGF not decreasing in figure 4D? The authors also say, as expected. If miR26a/b degrades RNA, would you not expect to see reduction in RNA levels as well?

Answer: It is well known that miRNAs suppress gene expression without causing mRNA degradation; animal miRNA inhibit gene expression at the post-transcriptional level.

16. As the authors used 0.4 um pore, all exosomes (<0.1 um) and secreted factors can pass through. How can the authors attribute the function to HGF alone? Can the authors block HGF by antibodies? Or block the receptors in the recipient cells? In this form, there are too many factors and hence HGF alone is not convincing.

Answer: In Figure 6D, HGF secretion was significantly inhibited using shRNA. And down-regulation of liver HGF clearly suppressed cell proliferation and migration (Figure 5E-G).

17. Is the results similar to other GC cell models as well?

Answer: Cell co-culture is widely used to determine the interaction between different cells. Previous study showed that SGC7901 can promote cell growth and migration of HUVEC cells.

Minor points

1. Second paragraph in introduction starts with "Extracellular vesicles".

Answer: "Extracellular vesicles" is replaced by "exosome" in the second paragraph.

2. Figure legend says "Photos" of exosomes. This is electron micrograph. Please see other

exosome articles on how these are referred previously.

Answer: “photos” is replaced by “Electron-microscope scanning of exosomes”.

3. Results (line 194), Figure 2D should be 1D.

Answer: This has been corrected.

4. Figure 1B, spelling mistake on exosomes.

Answer: Many thanks. The label of “exosome” has been corrected.

Reviewers' comments:

Reviewer #1 (Remarks to the Author):

The authors responded to, and were able to address previous concerns with additional data. However, some issues remain unresolved.

1. The liver cells producing HGF in response to exosomal EGFR have not been characterized in the in vitro model or in vivo. The authors provide WB data that CD44 and F4/80 are present in the cultured liver cells. CD44 is expressed on many cell types and therefore not helpful in this context. Also, in Fig 2E the authors show co-localization of exosomal EGFR with E-cadherin, clearly a marker of epithelial and not stromal cells. So are the exosomes taken up mainly by hepatocytes and if yes, are these cells producing HGF? Immunocytochemistry performed directly on the cultured liver cells can resolve some of these issues.

Please note that in the rebuttal letter the authors indicate use of CD166 and CD31 to characterize the liver cells, but no data are shown for either. Also, BM8 (designation of Ab) should be replaced with F4/80 (Kupffer cell marker) in the text.

2. In response to concerns regarding the experimental design shown in Fig 6 the authors responded "The lenti-viruses containing HGF shRNA by in situ multi-point injection. These lenti-viruses were injected 6 days prior to tumor implantation because it takes several days for lenti-viruses to up-regulate or down-regulate a specific gene in vivo....."

The end point of this experiment is liver metastasis and this was analyzed 60 days post tumor implantation. Metastasis is a late event relative to time of tumor implantation and this raises the question of how durable was the effect of lentivirus transduction and was it still effective weeks after the initial injections, at a time when tumor derived exosomes are reaching the liver and tumor cells also begin to disseminate. The authors show one WB with reduced HGF levels in livers of lentivirus-injected mice. It is not clear when this was performed relative to injection time. For the link between HGF and metastasis to be compelling, HGF levels should be monitored over time. Optimally, liver cell transduction by lentivirus should be confirmed by fluorescence microscopy.

3. Previously for Fig 7E."The authors should also provide IHC evidence for increased phospho-c-Met levels.in the metastases...".

This was not done. WB data on whole liver and liver metastases are shown but no quantitative data is provided and the differences between the treatment groups appear to be relatively minor. These data are crucial to the authors' conclusions and should be solid.

Additional comments:

A quantitative analysis should be provided for all WB data and the number of repeats indicated.

Reviewer #3 (Remarks to the Author):

The authors have addressed some parts of the issues raised. It will be great if the authors include these following issues in the discussion.

"And these EGFR-absent exosomes lost the function of promoting HGF expression in primary liver cells (Figure 3A). "

By knocking down EGFR, how sure are the authors that exosomes from EGFR knock down cells will

have same protein and RNA content (as wild type). EGFR knock down will definitely change the content of exosomes. Can the authors at least mention this in discussion?

Also, this reviewer is still not convinced with the exosome EGFR to liver cell membrane localization. Needs more data to be convincing. At this stage, it looks preliminary. Also, cetuximab or exosome blocking can give more concrete data.

Reviewer #4 (Remarks to the Author):

The authors addressed most of the comments of reviewer 2. However, there are a couple of points raised by the reviewer that still need to be clarified by the authors:

-Comment: It is unclear if transport of additional EGFR by EVs can significantly increase total EGFR levels in these cells, which would lead to a qualitatively different signaling outcome.

The authors replied "We checked the effects of SGC exosomes on the expression of liver EGFR (Figure 3A). It was showed that EGFR was significantly up-regulated by exosomes from cancer cells." However, in figure 3A the effects of exosomes on HGF and not EGFR are shown.

-Comment: Fig. 6. Evidence that EV-containing EGFR is responsible for the observed effect is lacking.

The authors did not reply.

Reviewer #1 (Remarks to the Author):

1. The liver cells producing HGF in response to exosomal EGFR have not been characterized in the in vitro model or in vivo. The authors provide WB data that CD44 and F4/80 are present in the cultured liver cells. CD44 is expressed on many cell types and therefore not helpful in this context. Also, in Fig 2E the authors show co-localization of exosomal EGFR with E-cadherin, clearly a marker of epithelial and not stromal cells. So are the exosomes taken up mainly by hepatocytes and if yes, are these cells producing HGF? Immunocytochemistry performed directly on the cultured liver cells can resolve some of these issues. Please note that in the rebuttal letter the authors indicate use of CD166 and CD31 to characterize the liver cells, but no data are shown for either. Also, BM8 (designation of Ab) should be replaced with F4/80 (Kupffer cell marker) in the text

Answer: To make sure that exosomes containing EGFR can be taken up by liver stromal cells; we conducted immuno-fluorescence assay in primary liver cells. Desmin and α -SMA are widely used as the marker of hepatic stellate cells (HSCs), and F4/80 is known as the marker of Kupffer cells. These markers are proved to be co-expressed with EGFR-GFP delivered by exosomes (Figure 3C). Western blotting analysis confirmed that the primary liver cells contain HSCs and Kupffer cells (Figure 3A and 3B).

2. In response to concerns regarding the experimental design shown in Fig 6 the authors responded “The lenti-viruses containing HGF shRNA by in situ multi-point injection. These lenti-viruses were injected 6 days prior to tumor implantation because it takes several days for lenti-viruses to up-regulate or down-regulate a specific gene in vivo.....”

The end point of this experiment is liver metastasis and this was analyzed 60 days post tumor implantation. Metastasis is a late event relative to time of tumor implantation and this raises the question of how durable was the effect of lentivirus transduction and was it still effective weeks after the initial injections, at a time when tumor derived exosomes are reaching the liver and tumor cells also begin to disseminate. The authors show one WB with reduced HGF levels in livers of lentivirus-injected mice. It is not clear when this was performed relative to injection time. For the link between HGF and metastasis to be compelling, HGF levels should be monitored over time. Optimally, liver cell transduction by lentivirus should be confirmed by fluorescence microscopy.

Answer: Thanks to your good suggestion. Maybe we did not make it clear that mouse liver were obtained at 60th day post tumor implantation. HGF expression shown in Fig 6D suggested that these lenti-viruses still work in mouse liver when

mice were sacrificed.

3. Previously for Fig 7E.”The authors should also provide IHC evidence for increased phospho-c-Met levels.in the metastases...”.

This was not done. WB data on whole liver and liver metastases are shown but no quantitative data is provided and the differences between the treatment groups appear to be relatively minor.

These data are crucial to the authors' conclusions and should be solid.

Answer: Phospho-c-MET are now determined by IHC as well as western blotting analysis (Figure 8F and 8H). It is clearly shown that high levels of liver HGF promotes the activation of c-MET in liver metastases.

4. Additional comments:

A quantitative analysis should be provided for all WB data and the number of repeats indicated.

Answer: Quantitative analysis for WB is newly added in Fig 1C, 2B, 3B, 4A, 4C, 6B, 6H and 8H.

Reviewer #3 (Remarks to the Author):

1. The authors have addressed some parts of the issues raised. It will be great if the authors include these following issues in the discussion.

"And these EGFR-absent exosomes lost the function of promoting HGF expression in primary liver cells (Figure 3A). "

By knocking down EGFR, how sure are the authors that exosomes from EGFR knock down cells will have same protein and RNA content (as wild type). EGFR knock down will definitely change the content of exosomes. Can the authors at least mention this in discussion?

Also, this reviewer is still not convinced with the exosome EGFR to liver cell membrane localization. Needs more data to be convincing. At this stage, it looks preliminary. Also, cetuximab or exosome blocking can give more concrete data.

Answer: Thanks to your kind suggestions. The three issues have been added in the discussion.

Reviewer #4 (Remarks to the Author):

The authors addressed most of the comments of reviewer 2. However, there are a couple of points raised by the reviewer that still need to be clarified by the authors:

1. Comment: It is unclear if transport of additional EGFR by EVs can significantly increase total EGFR levels in these cells, which would lead to a qualitatively different signaling outcome. The authors replied “We checked the effects of SGC exosomes on the expression of liver EGFR (Figure 3A). It was shown that EGFR was significantly up-regulated by exosomes from cancer cells.” However, in figure 3A the effects of exosomes on HGF and not EGFR are shown.

Answer: Western blotting analysis of HGF in primary liver cells treated with SGC exosomes were newly added, and Both HGF and EGFR expression are shown in Fig 4A now.

2. Comment: Fig. 6. Evidence that EV-containing EGFR is responsible for the observed effect is lacking. The authors did not reply.

Answer: Effects of exosome-EGFR on the expression of liver HGF have been clearly demonstrated in vitro (Fig 4A). Moreover, EGFR-deleted exosomes lost the function of enhancing liver HGF expression (Fig 2B and 4A).

SGC exosome containing EGFR-GFP were injected into mice via tail vein, and we detected the EGFR-GFP in liver (Figure S2). It is shown that exosome-EGFR can be located in the cell membrane of liver in vivo, and liver HGF is up-regulated with EGFR transported by SGC exosomes (Fig 7D).

Therefore, exosome-containing EGFR is responsible for the observed in-vivo effect.

REVIEWERS' COMMENTS:

Reviewer #4 (Remarks to the Author):

The authors have addressed the points that I raised.

Reviewer #5 (Remarks to the Author):

I feel that this is an interesting paper and fits with Nature Communication. As the Editor suggested, I examined whether the authors addressed the original reviewer 1's concerns. I think that new data provided by authors have successfully addressed these concerns.

REVIEWERS' COMMENTS:

Reviewer #4 (Remarks to the Author):

The authors have addressed the points that I raised.

Answer: Thanks for your affirmation of our study.

Reviewer #5 (Remarks to the Author):

I feel that this is an interesting paper and fits with Nature Communication. As the Editor suggested, I examined whether the authors addressed the original reviewer 1's concerns. I think that new data provided by authors have successfully addressed these concerns.

Answer: Thank you so much for your support to our work.